# State-of-the-art efficiency determination of a wind turbine drivetrain on a nacelle test bench

Hongkun Zhang[1], Paula Weidinger[2], Christian Mester[3], Zihang Song[2], Marcel Heller[1], Alexander Dubowik[2], Bernd Tegtmeier[1], and Karin Eustorgi[1]

[1]Fraunhofer Institute for Wind Energy Systems IWES, 27572 Bremerhaven, Germany
[2]Physikalisch-Technische Bundesanstalt, 38116 Braunschweig, Germany
[3]Federal Institute of Metrology METAS, 3003 Bern-Wabern, Switzerland

**Correspondence:** Karin Eustorgi (karin.eustorgi@iwes.fraunhofer.de)

**Abstract.** The efficiency of wind turbine drivetrains is a topic of great interest for both the wind energy industry and the academic community. With the developing maturity of this technology and the increasing pressures to reduce costs, the importance of drivetrain efficiency has grown. However, measuring the mechanical input power and the electrical output power with sufficient accuracy is very challenging due to the high power level of the drivetrain. In the project known as WindEFCY, state-of-the-art measurement and calibration instruments are used to determine the drivetrain efficiency of a direct drive wind turbine on the nacelle test bench called DyNaLab. This paper discusses the test configuration applied for this work as well as the instrumentation of the measurement systems used. It further presents the results from two tests of different types to demonstrate the process of efficiency determination, the analysis of uncertainty and the consequent comparability of the tests. Within the paper's scope of study, an uncertainty level of approximately 0.7% is achievable when measuring drivetrain efficiency. Details and recommendations concerning data processing and uncertainty analysis are also given in the paper.

## 1 Introduction

To further reduce the levelised cost of energy (LCOE), increasing the rated power of a single wind turbine is still a common and effective approach that is actively pursued by the wind energy industry. When developing larger wind turbines, it is of key interest to maximise the efficiency of wind energy utilisation. This efficiency, however, is not a single and constant parameter. Wind turbines operate in a wide working range that at most times deviates from the rated power and speed. Moreover, wind turbines are constantly subjected to stochastic wind conditions that directly or indirectly influence efficiency. The efficiency property of a wind turbine therefore has to be determined across the entire working range and under different conditions. The determined efficiency property provides an important basis for turbine optimisation.

The overall efficiency of the wind turbine consists of the aerodynamic efficiency of the rotor and the efficiency of the drivetrain. The drivetrain efficiency is affected by a number of factors that have to be considered in the design of the turbine. These factors include the setting and functionality of the cooling system, the structural deformation (especially the air gap change for direct drive turbines) due to external loads and temperature change, as well as control strategies of the generator and converter. In order to validate and optimise the turbine design, the influence of these factors on the drivetrain's efficiency needs

to be determined both qualitatively and quantitatively. To do so, it is necessary to determine the efficiency with a high level of accuracy and with measurements traceable to national standards according to metrological rules (Weidinger et al., 2021). Traceability to national standards as well as the accuracy and precision defined by the calibration, which is a comparison of the measurements on hand with national standards, are of great importance for optimising efficiency. A performance comparison of different drivetrain components or working conditions is only meaningful if the measurement methods and conditions are sufficiently accurate to be able to compare the measurement results with each other.

The best place to determine the drivetrain efficiency of a wind turbine is on a nacelle test bench, where the design mechanical load cases and electrical grid conditions can be easily produced and replicated. This said, determining efficiency with sufficient accuracy is still very challenging even on nacelle test benches. Major reasons for this are the lack of traceability to national standards in both the mechanical and the electrical fields. Metrological traceability is defined as "the property of a measurement result whereby the result can be related to a reference through a documented unbroken chain of calibrations, each contributing to the measurement uncertainty." (Vim, 2004) It ensures the accuracy of a measurement described with the measurement uncertainty that is given by the calibration chain. On nacelle test benches, calibration of the torque measurement is especially challenging due to the high torque levels and the absence of torque standards above 1.1 MN·m (Foyer et al., 2019). A few approaches have been suggested to avoid the need for torque measurement when determining efficiency on nacelle test benches, including the calorimeter method (Pagitsch et al., 2016) and the modified back-to-back method (Zhang and Neshati, 2018). Nevertheless, the most reliably reproducible method of efficiency determination that is traceable to national standards is still the "direct" method, i.e., measuring the input and output power directly using state-of-the-art equipment. In this case, a 5 MN·m torque transducer including a device for rotational speed measurement owned by Physikalisch-Technische Bundesanstalt (PTB, the German National Metrology Institute) (Weidinger et al., 2017) was used to trace the mechanical power measurement. Additionally, a reference power measuring system (RPMS) calibrated by the Swiss National Metrology Institute METAS together with the PTB was used to trace the electrical power measurement to national standards.

## 2 Background

The WindEFCY project provided the opportunity to determine the efficiency of a wind turbine drivetrain with traceable measurement of both the mechanical input power and the electrical output power. The wind turbine drivetrain was tested on the 10 MW DyNaLab nacelle test bench of Fraunhofer IWES in Bremerhaven, Germany. During the test campaign, the 5 MN·m torque transducer (also known as the torque transfer standard, TTS), as well as other mechanical and electrical sensors were used to produce traceable measurements of the input and output powers. The TTS is linked to a primary national torque standard via calibration and transfers the highly accurate and precise torque measurement into industrial applications; here it serves as a reference standard to ensure the accuracy and traceability in torque measurement of the test bench's own torque transducer.

The WindEFCY project, officially titled "Traceable mechanical and electrical power measurement for efficiency determination of wind turbines", was a collaborative research initiative across disciplines, such as mechanical and electrical power measurement

and wind turbine test bench operation, under the European Metrology Programme for Innovation and Research (EMPIR). Aim of the project was the development and validation of standardised test methods for the efficiency determination of wind turbine drivetrains and their components on test benches in a reliable, reproducible, and comparable way for quality assurance. This required several calibration processes and adequate standards. In a calibration process, the unknown measuring instrument is compared with a known standard provided by National Metrology Institutes, and the accuracy of the unknown measuring instrument is specified within certain tolerances, the measurement uncertainty (MU). Without traceability of the measurands, the accuracy and precision of the measurands are neither known nor reliable. Consequently, the measured quantities cannot be compared with information in data sheets or measurement data acquired by other measuring equipment or in other test benches. To optimise the efficiency of wind turbine drivetrains, where already a good efficiency is achieved, measurements with high accuracy and precision are essential. Within the WindEFCY project, it was shown that – especially in the field of torque measurement in the mega-newton metre range – traceability is obligatory, as the torque measurement discrepancy in test benches occured to be ± 5%. For discrepancies of this magnitude, it is important that the calibration not only determines the MU, i.e. the range within which the true value of the measured quantity is likely to lie, but also an adjustment to the torque measurement is made to align it with the standard.

The efficiency behaviour of the turbine drivetrain was determined on numerous working points at 5 MN·m and under different conditions. In addition to aiding the efficiency determination, the availability of the 5 MN·m TTS also offered a chance to calibrate the test bench's own torque transducer, which is instrumented on the shaft adapter connecting the test bench and the device under test (DUT), as shown in Figure 1. For the calibration, a number of calibration profiles were also carried out during the test campaign. Since the 5 MN·m TTS is not designed for high levels of non-torque loads (also known as parasitic loads in some publications), the calibrated test bench transducer could be used instead in future tests with high non-torque loads.

## 3 Test layout for efficiency determination

The layout of the complete test setup is depicted schematically in Figure 1. Two motors of the test bench are connected in tandem to provide the driving torque. A load application unit (LAU) can be used to generate the designed non-torque loads with a hexapod driven by hydraulics. A coupling is placed between the LAU and the motors to prevent non-torque loads being transferred backwards to the motors. The non-torque loads are transferred via a main bearing from the hexapod to the output shaft that also carries the torque. A combination of loads in six degrees of freedom can be applied to the DUT through the flange of the output shaft. To connect the test bench with the DUT, a shaft adapter is used to fit the flanges on both sides. This adapter (yellow in Figure 1) is also used as a robust way to measure loads, i.e. torque, bending moments, and axial as well as lateral forces, directly in front of the DUT. For the WindEFCY test campaign, the 5 MN·m TTS from PTB (red in Figure 1) was additionally integrated into the setup between the shaft adapter and the DUT with the help of specially designed adaptation structures. To protect the 5 MN·m TTS, the applied non-torque load was controlled to the minimum during the tests.

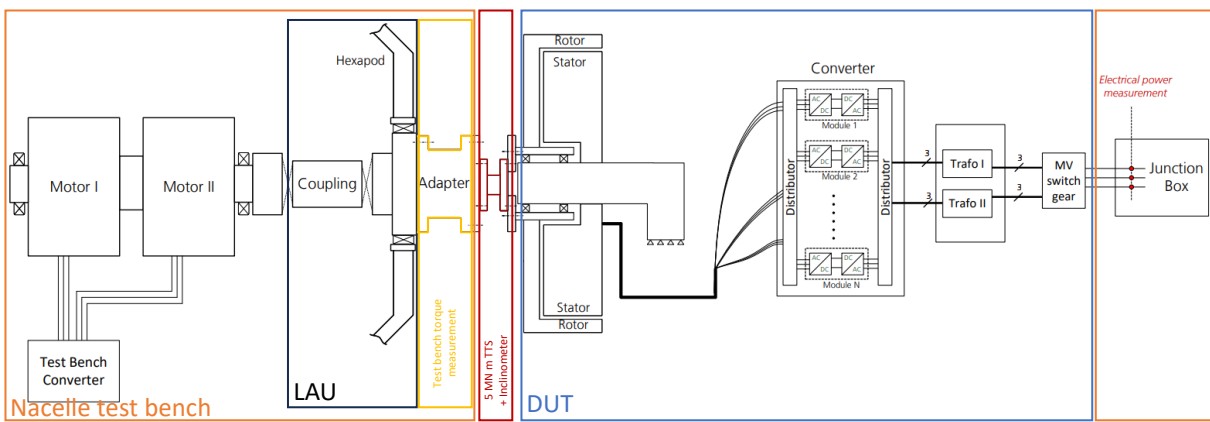

**Figure 1.** Layout of the test configuration with the test bench on the left, the DUT in the middle and the junction box to the grid on the right. The mechanical power is measured between test bench and the DUT with the 5 MN·m TTS together with a inclinometer. The electrical power is measured in the junction box.

The non-rotating part of the DUT is fixed to the base of the test bench. The DUT's generator is electrically connected to a full power converter, which is in turn connected to the transformer. Via a switch gear, the transformer is connected to the medium voltage inside a junction box. For efficiency determination, the mechanical power is measured by the 5 MN·m TTS and the electrical power is measured by the electrical power measurement system (EPMS) in the junction box. The efficiency is then determined for all the components in between, including the generator, the converter and the transformer.

## 4   Measurement of mechanical input power

The mechanical input power $P_{\mathrm{mech}}$ of the DUT is a function of the input torque $M$ and the rotational speed $n$ at the interface of the DUT, as shown in Equation 1. It is very important that the torque and rotational speed being measured at the same position as the power is calculated from both of them. To this end, the 5 MN·m TTS was augmented by an inclinometer to measure rotational speed. In this chapter, important details of both the torque and the speed measurement are presented.

$$P_{\mathrm{mech}} = M \cdot n \tag{1}$$

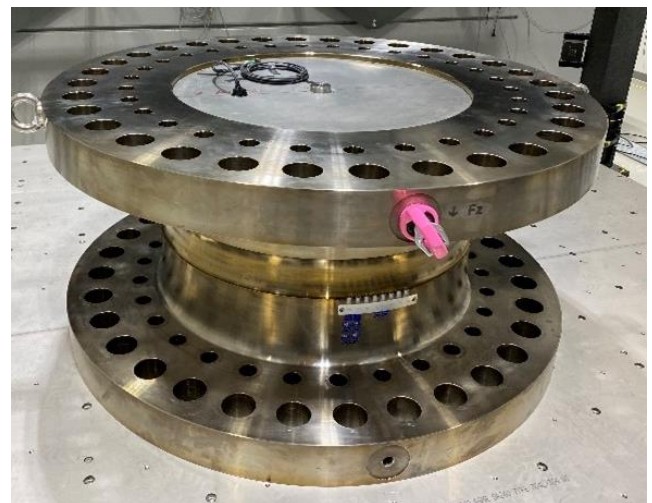 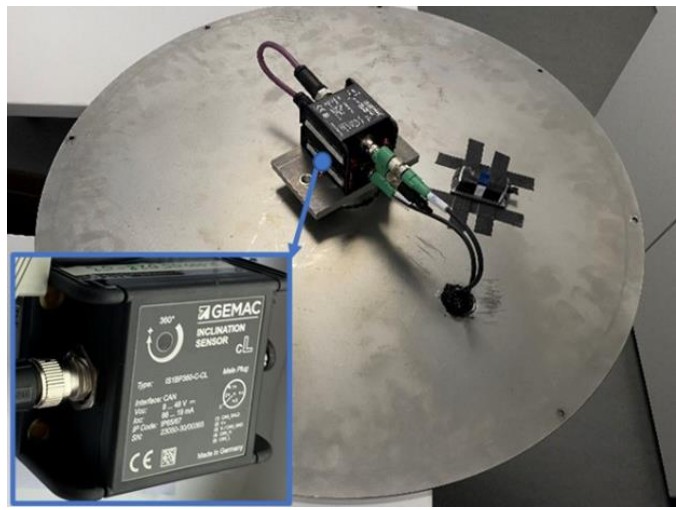

**Figure 2.** The 5 MN·m TTS manufactured by HBM (left image). It is a hollow-shaft transducer with flanges to be mounted in test benches. In the left image, the hollow-shaft is covered by a cover plate. On the inside of this cover and placed centred is the inclinometer stack (zoom in right image). Moreover, a small sensor to log temperature and humidity inside the TTS is taped to the inner side of the cover plate (right image, black tape). Taken from (Song et al., 2022).

### 4.1 Torque measurement with the 5 MN·m TTS

To meet the needs of the wind energy industry, a 5 MN·m TTS manufactured by the company Hottinger Baldwin Messtechnik (HBM) was acquired by PTB especially for the use on nacelle test benches. It is pictured in Figure 2 (left). The TTS is equipped with strain gauges interconnected in a Wheatstone bridge circuit to measure torque up to 5 MN·m. Additionally, it can also measure bending moments, axial and shear forces, but only to a lower level, and these additional measurements are not traceable to national standards. The TTS is statically calibrated using PTB's 1.1 MN·m torque standard machine in order to establish a relationship between the transducer's output signal $S_{\text{TTS}}$ (in mV/V) and the applied torque $M$ (in kN·m). This was done according to the German torque calibration standard DIN 51309, but only up to 1.1 MN·m due to the lack of suitable torque standard machines.

DIN 51309 stipulates which parameters must be determined in addition to the calibration result in order to calculate the MU. These include reproducibility $b$, repeatability $b'$, resolution $r$, zero point deviation $f_0$ and regression or display deviation $f_a$. Due to the hysteresis behaviour $h$ of torque transducers in case of load direction changes, two cases are distinguished when evaluating the sensor behaviour. In Case I, only increasing torque is measured and a linear or cubic regression curve through the origin is calculated, which is used for the future signal display. As there is no load change in this case, the hysteresis behaviour does not need to be taken into account in the MU. In Case II, the sequence of the load, whether increasing or decreasing, is not known. Accordingly, a linear regression curve is calculated based on combined calibration data of both increasing and decreasing torque. The hysteresis of the sensor occurring with alternating load contributes to the overall MU.

With regard to its application on the test bench, where both increasing and decreasing torque can occur, a linear regression curve combined for increasing and decreasing torque was determined for the calibration of the 1.1 MN·m TTS in accordance with Case II in DIN 51309. Using the sensitivity and the TTS's signal $S_\text{TTS}$, the torque $M$ can be calculated:

$$M = 3850\,\text{kN} \cdot \text{m} \cdot \left(\frac{\text{mV}}{\text{V}}\right)^{-1} \cdot S_\text{TTS} \tag{2}$$

Due to the lack of possible calibration devices above 1.1 MN·m, the behaviour of the TTS up to 5 MN·m can only be predicted. With a relative linearity deviation of $0.7 \times 10^{-4}$ at 1.1 MN·m and a relative hysteresis of $< 6.2 \times 10^{-4}$ in the measurement range up to 1.1 MN·m, the TTS exhibits exceptionally linear behaviour. The assumption of partial range sensitivity for full range measurements is supported by the stability of the sensitivity in further partial measurement ranges (8%, 12% and 16%) with relative deviations of $8 \times 10^{-8}$ noticeably below the overall MU of $8 \times 10^{-4}$.

The use of a partial range sensitivity for a full range measurement was validated using partial and full range measurements on a very well characterised 20 kN·m torque transfer standard. The relative deviation of the partial range sensitivity from the full range sensitivity was $3 \times 10^{-6}$, which was below the smallest possible overall measurement uncertainty for Case I with cubic regression curve.

In addition to the sensitivity, the overall MU is the result of a calibration. Therefore, a MU must be specified for the full measurement range of the TTS on top of the predicted sensitivity. The MU is also a prediction, which should be treated carefully as such, and does not replace a real calibration, where this is possible. A weighted extrapolation method (Weidinger et al., 2023) was used to predict the overall MU for the full measurement range. The maximum overall MU of the partial range calibration is multiplied by a prediction factor $f_\text{w}$ to account for the uncertainty of the extrapolation itself. This factor is the sum of a scaling factor $f_\text{s}$ and the classification criteria of the partial range calibration according to DIN 51309.

Besides determining the MU, the classification of measuring devices is a way of presenting the accuracy of the calibrated measuring device in a standardised way that can be recognised at a glance. Classification requires a combination of specific criteria to be met. The classification criteria according to DIN 51309 are: relative reproducibility $b_\text{rel}$, relative repeatability $b'_\text{rel}$, relative zero point deviation $f_\text{0,rel}$, relative hysteresis $h_\text{rel}$, and relative regression deviation $f_\text{a,rel}$, as well as the resolution of the TTS at the smallest calibrated lower range value $M_\text{A}$ and the relative MU of the torque standard machine $W_\text{C}$.

$$f_\text{w} = f_\text{s} + b_\text{rel} + b'_\text{rel} + f_\text{0,rel} + h_\text{rel} + f_\text{a,rel} + M_\text{A} + W_\text{C}. \tag{3}$$

The scaling factor $f_\text{s}$ is the ratio of maximum extrapolated torque $M_\text{ex}$ to the maximum calibrated torque $M_\text{C}$ and, therefore, weights the uncertainty higher with increasing extrapolation:

$$f_\text{s} = \frac{M_\text{ex}}{M_\text{C}}. \tag{4}$$

Due to the relative MU of the torque standard machine of $8 \times 10^{-4}$, the class of the TTS is 0.5. For the extrapolated range, the corresponding scaling and weighting factors and the extrapolated MU are shown in Table 1.

**Table 1.** Scaling and weighting factor for the 5 MN·m TTS calibrated in the sub-range up to 1.1 MN·m and extrapolated relative expanded MU for the range between 1.5 MN·m and 5 MN·m

| Steps | $f_s$ | $f_w$ | Extrapolated relative expanded MU / % |
|---|---|---|---|
| 0 | - | - | - |
| 1500 | 0.11 | 3.2 | 0.27 |
| 2000 | 0.15 | 3.7 | 0.3 |
| 2500 | 0.19 | 4.1 | 0.34 |
| 3000 | 0.23 | 4.6 | 0.38 |
| 3500 | 0.26 | 5 | 0.42 |
| 4000 | 0.3 | 5.5 | 0.45 |
| 4500 | 0.34 | 5.9 | 0.49 |
| 5000 | 0.38 | 6.4 | 0.53 |

Weighted extrapolation approach

### 4.2 Measurement of rotational speed with the mechanical power transfer standard

The first challenge in establishing a transfer standard for measuring rotational speed in nacelle test benches stems from the difficulty of installing the encoder stator in very close proximity to the rotating shaft. This problem is attributed to the towering height of the rotor hub and the absence of rigid structures. To address this, a stator-free method for measuring rotational speed has been developed using a specially chosen inclinometer. This inclinometer, which functions as a microelectromechanical system (MEMS), contains two perpendicular accelerometers that determine inclination relative to gravity. Placed at the centre of the drivetrain, the inclinometer measures the angular position ($\phi$) of the rotating shaft with respect to gravity. The average rotational speed ($n$) is then calculated based on the change in angle ($\Delta\phi = \phi_2 - \phi_1$) and the elapsed time ($\Delta t$), following the formula:

$$n = \frac{\Delta\phi}{\Delta t} \cdot \frac{60}{360°} \tag{5}$$

The inclinometer's calibration was performed at the length and angle laboratory at PTB and yielded an expanded MU (with coverage factor $k = 2$) of 0.014° under static conditions using a 0.22 Hz Bessel lowpass filter. Utilising its stator-free characteristic, the inclinometer was mounted on the inner side of the TTS cover plate at the centre part, as depicted in Figure 2. The overall MU ($u_n$) of rotational speed is influenced by uncertainties in angle ($u_\phi$) and time ($u_t$), and is determined by the equation:

$$u_n^2 = \left(\frac{\partial n}{\partial \phi_1} \cdot u_\phi\right)^2 + \left(\frac{\partial n}{\partial \phi_2} \cdot u_\phi\right)^2 + \left(\frac{\partial n}{\partial \Delta t} \cdot u_t\right)^2 \tag{6}$$

The standard uncertainties of the angle measurement $u_\phi$ and the time measurement $u_\mathrm{t}$ contribute to the total standard uncertainty $u_\mathrm{n}$ of the rotational speed measurement:

$$u_\mathrm{n} = \frac{60}{360°} \cdot \sqrt{2\left(\frac{u_\phi}{\Delta t}\right)^2 + \left(\frac{\phi_2 - \phi_1}{\Delta t^2} \cdot u_\mathrm{t}\right)^2} \tag{7}$$

It is obvious that $u_\mathrm{n}$ decreases as the time interval $\Delta t$ increases, thereby reducing uncertainty. To ensure synchronised measurements, the rotational speed was always measured over the same interval as the torque measurement.

Incorporating additional uncertainties arising from mounting misalignments, eccentricity, dynamic effects, and data evaluation processes, the total relative expanded MU for the rotational speed measurement on the nacelle test bench was calculated as 0.02%. Using the aforementioned inclinometer, which was developed as a transfer standard for rotational speed and integrated with the 5 MN·m TTS on the nacelle test bench, a traceability chain for mechanical power measurement is established (Weidinger, 2023).

## 5  Measurement of electrical output power

Efficiency is the ratio of useful electrical output power $P_\mathrm{elec}$ converted from the available mechanical input power. The electrical output power is the sum of the power of all three phases $P_{1-3}$, whereas the power per phase $P$ is determined as the product of transient current $i_{1-3}$ and voltage $u_{1-3}$:

$$P_\mathrm{elec} = P_\mathrm{elec,phase1} + P_\mathrm{elec,phase2} + P_\mathrm{elec,phase3}, \tag{8}$$

$$P_\mathrm{elec,phase1-3} = \frac{1}{T} \int_0^T u_{1-3}(t) \cdot i_{1-3}(t) dt. \tag{9}$$

Since the electrical output power is intended to be fed to the electricity grid, only the electrical power at grid frequency is useful. Power quality phenomena such as other spectral components are relevant since they influence, for instance, the stability of the grid. These are also studied, but since they are not considered as "useful output" for determining efficiency, the uncertainty requirements are much less stringent. The setup of the electrical gauges in the DyNaLab nacelle test bench is shown in simplified form in Figure 3.

In nacelle test benches, electrical power is usually measured using an electrical power measurement system (EPMS) integrated into the test bench's data acquisition (DAQ) system. This system is optimised for convenience and versatility, not for minimum uncertainty. Since it is integrated into the nacelle test bench, sending this system to calibration laboratories is difficult and time-consuming. For this reason, METAS and PTB calibrated a reference power measuring system (RPMS) for use in test benches. This system is used to determine the efficiency and to calibrate the EPMS measurement chain of the test bench on site.

The RPMS is based on commercial off-the-shelf components such as the LMG671 power analyser and the DL 2000ID current sensors (Figure 3). As the planned reference voltage divider HST12-3 could not be used due to the risk of over-voltage and to test hall safety regulations, the more robust HILO voltage dividers were employed instead, and an extensive recalibration of the HILO sensors had to be carried out at PTB. The over-voltage risk stemmed from the fact that the DUT had to be connected to the medium-voltage grid of the local grid operator instead of to the grid simulator of the test bench. This had

the disadvantage that in the event of a fault in the medium-voltage grid, high over-voltages could occur due to ground faults or lightning strikes.

The power analyser was used for measurements at the primary side of the transformer. The power analysers are calibrated using a modular primary power standard (Mester, 2021). Depending on the currents and voltages to be measured, transducers can be used to reduce the currents and voltages to levels that can be measured with the power analyser. These reference

transducers are calibrated with an uncertainty of 300 $\mu$V/V and 30 $\mu$A/A at power frequency.

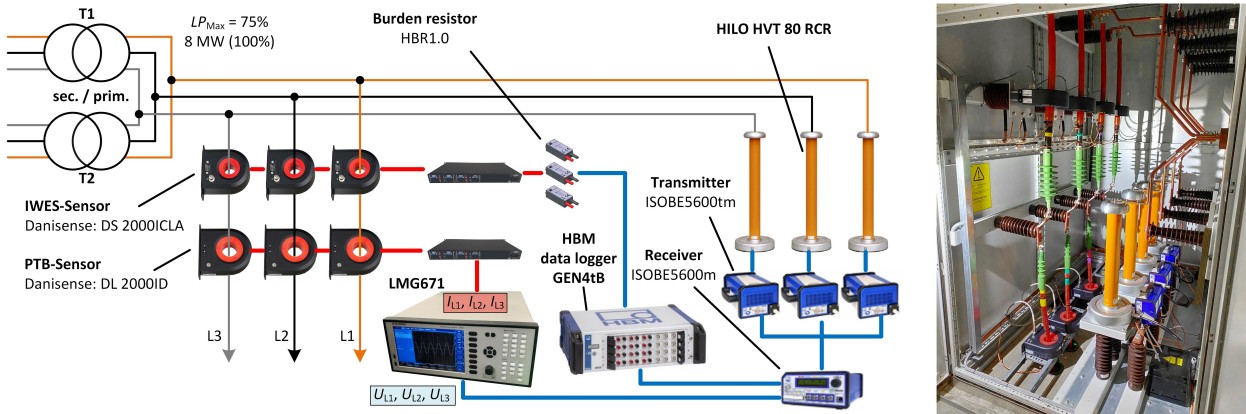

**Figure 3.** Schematic setup (left) and picutre (right) of the RPMS and the EPMS in the junction box of the DyNaLab nacelle test bench.

Figure 3 shows that the current measurement chain of the EPMS, consisting of current transformers of type DS 2000 ICLA, burden resistors of type HBR1.0, and the HBM data logger GEN4tB with the current measurement card GN8103B, which could be calibrated on site. The MU for the measurement chain is 0.01% and is valid for various load cases. A statement about the long-term stability cannot be made here.

Due to the over-voltage risk mentioned above, calibration of the HILO sensors on site was not possible, as the over-voltages could have damaged the voltage reference sensor of the RPMS. To calibrate the HILO voltage sensors, the entire measurement chain, consisting of voltage divider, connection cable, transmitter, fibre optic cable, and receiver (Figure 4), was shipped to and calibrated in the laboratory at PTB following the test campaign. The calibration was performed as a comparison measurement against a PTB standard. Due to a high position dependence of the voltage divider to other voltage dividers, the calibration

resulted in an expanded MU ($k = 2$) of 0.8% at power frequency, with the standard uncertainty being 0.4%.

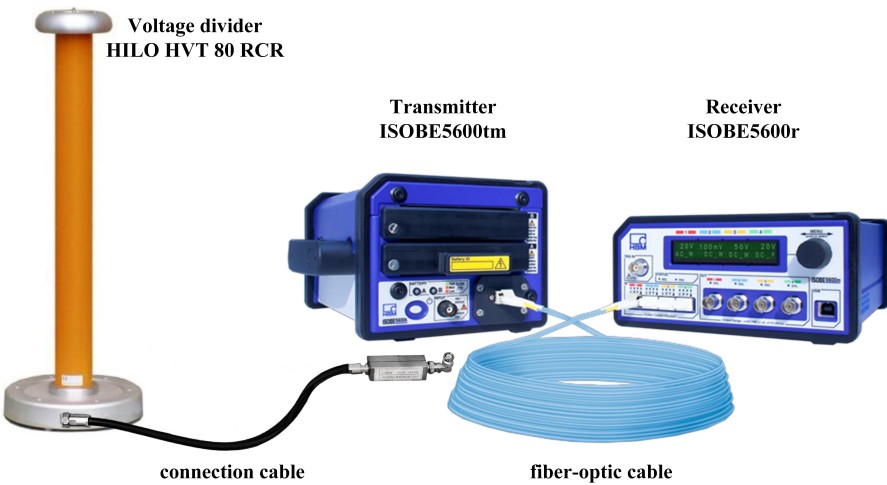

**Figure 4.** Measuring chain of the EPMS voltage path. Source: https://www.hbm.com/en/2343/isobe5600-isolation-system-standalone-transient-recorder, last access: $26^{th}$ July 2024, modified by the authors with permission of Hottinger Brüel & Kjaer GmbH

As shown in Table 2, the uncertainty in the voltage measurement $U_{1-3}$ plays a dominant role in the overall MU of the electrical power. Thanks to the state-of-the-art sensors and measurement system, the current $I_{1-3}$ can be measured with an extremely small uncertainty. Additionally, since the power factor $\lambda_{1-3}$ of the turbine is kept at 1 during the test, the uncertainty in $\lambda$ due to the phase errors of voltage and current sensors is negligible. The uncertainties of the three voltage measurements for the three phases are regarded as independent after the calibration and correction. As a result, the total power of the three phases has a smaller relative uncertainty compared to the power $P_{1-3}$ of each individual phase.

**Table 2.** Relative MUs for the single components of the electrical power measurement, meaning voltage, current, power factor, and electrical power for the three phases, as well as combined relative MU for the overall electrical power.

| $U_1, U_2, U_3$ | $I_1, I_2, I_3$ | $\lambda_1, \lambda_2, \lambda_3$ | $P_1, P_2, P_3$ | $P_{\text{elec}}$ |
|---|---|---|---|---|
| 0.40% | 0.01% | - | 0.40% | 0.23% |

For the determination of efficiency, the electrical power and mechanical power measurements need to be synchronised. While the mechanical power measurement system is synchronised to UTC (coordinated universal time) using IRIG-B signal, the chosen RPMS model cannot be synchronised to an external time reference other than by manually setting the time like on a wristwatch. Mechanical imperfections of the nacelle cause a pattern of mechanical power with a period of one revolution. Since the electrical power shows the same pattern, the electrical power measurement is synchronised in a post-processing stage using the cross correlation of the two power measurements.

## 6 Test results and analysis

During the test campaign, numerous tests serving various purposes were carried out. Because the measurement range of the
225 TTS (being 5 MN·m) is smaller than the rated torque of the DUT, all tests were carried out with the turbine operating below the rated power. The results of two tests are presented in this paper to demonstrate the method of efficiency determination. In both tests, the torque was held stable around the 5 MN·m level to utilise the maximum capacity of the transducer. In the first test, the rotational speed was kept constant to be able to analyse the long-term behaviour of the DUT and the temperature influence on the efficiency, while in the second test the rotational speed followed an operational curve in a stepwise manner upwards.
For each test, the mechanical input power and the electrical output power were calculated to determine the efficiency. The uncertainty analysis was carried out for the first test with uncertainty budgets of the raw measurements determined according to the propagation principle. The turbine under test was a customer device in development mode. Since the rated rotational speed is a key parameter for a commercial wind turbine, it was agreed to be normalised, which consequently leads to the normalisation of the mechanical and electrical powers in this paper.

### 6.1 Results from warm-up test

In the first test, the turbine was operated at a fixed working point for relatively long periods of time, as shown in Figure 5. This is named the warm-up test and is designed to study the change of temperature and hence the drivetrain efficiency over the course of long-term operation.

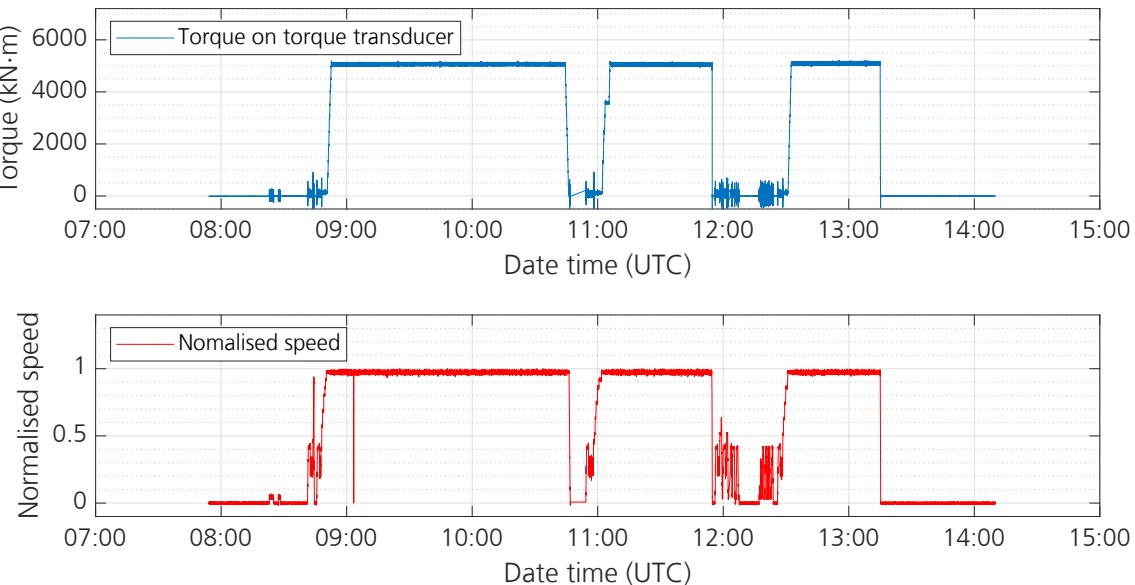

**Figure 5.** Actual test progress of the warm-up test: torque (upper part of the figure), measured by PTB's TTS, and rotational speed (lower part of the figure) measured by the rotary encoder of the test bench plotted against time.

In Figure 6 (upper plot), the mechanical and electrical power values are shown in the same plot for better comparison. The efficiency change over the course of operation time is visualised in the lower plot of the same figure. For better comparison, the mechanical and electrical power values are shown in the same figure. The general trend of efficiency drop with progressing operation time can be clearly seen. While the electrical power output is kept constant by means of the control strategy, the mechanical power input increases slowly. Each point in the upper graph represents the mean value of a 10-revolution power measurement; each point in the lower graph that of the corresponding efficiency result. The 10-revolution mean values allow better visualisation of the change trend.

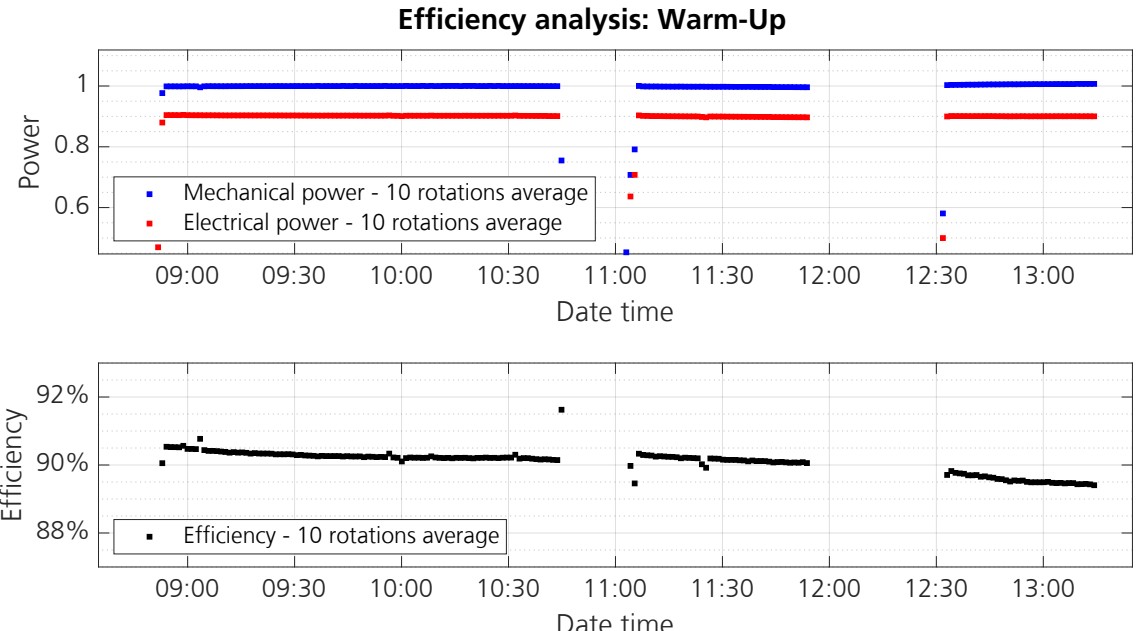

**Figure 6.** Efficiency determination for the warm-up test where both mechanical and electrical powers are depicted in the upper part of the figure and the efficiency, being calculated as the ratio of electrical output power and mechanical input power, is shown in the lower part of the figure. All variables are plotted as mean data points averaged over 10 revolutions.

For the reasons listed below, it is difficult, but also not necessary, to measure the instant efficiency at a specific time point. It makes more sense to measure the "mean" efficiency of the drivetrain for one or more revolutions.

– The drivetrain has notable inertia and can store and emit energy as the rotational speed fluctuates.

– The torsional vibration of the drivetrain as well as the speed control strategy cause ripples in the rotational speed and consequently in all other variables measured.

– The performance of the turbine generator, including its efficiency, is dependent on the air gap distribution between the rotor and stator along the circumference, which varies with the angular position of the rotor.

– The electrical power measurement is only carried out once per second. The power analyser can calculate the mean power within each second very accurately, but the power between any two outputs needs to be interpolated.

The deviation of the determined efficiency based on a 1-revolution averaged measurement can be calculated provided sufficient data is available. As an example, Figure 7 shows two versions of efficiency determination based on measurement data gathered over 100 revolutions; the upper part of the figure shows results of a 1-revolution average $\eta_1$, with the standard deviation of the 100 points being 0.13%. In the lower part of the figure, the efficiency was determined using measurement data being averaged over 10 revolutions, $\eta_{10}$, resulting in 10 points in the plot. The standard deviation of these 10 points is 0.019%.

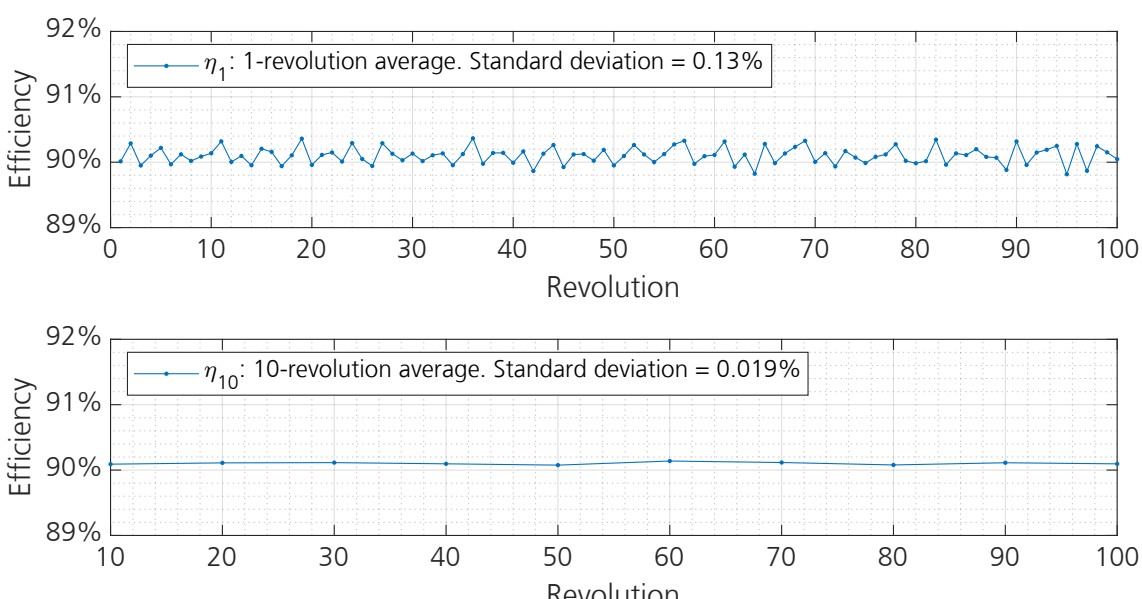

**Figure 7.** Standard deviation of 1-revolution and 10-revolution averaged efficiency. The very small drift of efficiency at different revolutions is compensated by a detrend operation in Matlab.

The standard deviations of $\eta_1$ and $\eta_{10}$ are denoted as $\sigma_{\eta_1}$ and $\sigma_{\eta_{10}}$. A comparison between the two is given in Table 3. According to the GUM guideline (JCGM, 2008), $\sigma_{\eta_{10}}$ would be equal to $\sigma_{\eta_1}$ divided by $\sqrt{10}$ if the uncertainties of the determined $\eta_1$ values are independent from each other. It should be noted, however, that $\sigma_{\eta_{10}}$ is much smaller than $\sigma_{\eta_1}/\sqrt{10}$, as shown in Table 3.

**Table 3.** Comparison of the standard deviations for efficiency $\eta_1$ and $\eta_{10}$.

| $\sigma_{\eta_1}$ | $\sigma_{\eta_1}/\sqrt{10}$ | $\sigma_{\eta_{10}}$ |
|---|---|---|
| 0.13% | 0.04% | 0.019% |

This indicates that in this case the determined values of efficiency with single revolution measurement $\eta_1$ are not independent
in terms of MU. Nevertheless, since $\sigma_{\eta_1}/\sqrt{10}$ gives a larger uncertainty than $\sigma_{\eta_{10}}$, and $\sigma_{\eta_1}$ needs a much smaller period of

measurement to be calculated than $\sigma_{\eta_{10}}$ , it remains meaningful to use $\sigma_{\eta_1}/\sqrt{10}$ as a conservative estimation (Equation 10) of $\sigma_{\eta_{10}}$ if the measurement period or the number of revolutions is limited.

$$\sigma^*_{\eta_{10}} = \sigma_{\eta_1}/\sqrt{10} \tag{10}$$

The detailed uncertainty budget for the determined efficiency with the averaged measurement of 10 revolutions is shown in Table 4. The left side of the table presents the uncertainty contributions from the measured electrical and mechanical variables. These are used to determine the efficiency uncertainty associated solely with the measurement chains and denoted as $u_{\eta,\mathrm{meas}}$. On the right side of the table, the uncertainty associated with the instability in the efficiency is indicated, with the standard deviation adopted as the standard uncertainty $u_{\eta_{10},\mathrm{ins}}$. In this case, the 10-revolution average is used. If the efficiency is determined with the average of a different number of revolutions, the corresponding standard deviation should be used.

**Table 4.** Overall uncertainty budget of the determined drivetrain efficiency

| Current | Voltage | Torque | Speed | |
|---|---|---|---|---|
| $u_\mathrm{I} = 0.01\%$ | $u_\mathrm{V} = 0.4\%$ | $u_\mathrm{M} = 0.27\%$ | $u_\mathrm{n} = 0.01\%$ | |
| Electrical power | | Mechanical power | | Instability in efficiency |
| $u_{\mathrm{P_{elec}}} = 0.23\%$ | | $u_{\mathrm{P_{mech}}} = 0.27\%$ | | |
| Uncertainty caused by measurement chains | | | | |
| $u_{\eta,\mathrm{meas}} = 0.35\%$ | | | | $u_{\eta_{10},\mathrm{ins}} = \sigma^*_{\eta_{10}} = 0.041\%$ |
| Overall | | | | |
| $u_{\eta_{10}} = 0.35\%$, expanded MU $U_{\eta_{10}} = 0.70\%$ $(k = 2)$ | | | | |

Combining the contributions of measurement chains and the instability yields the overall uncertainty in efficiency shown at the bottom of the table. Obviously, the contribution of instability in this case plays only a negligible role in the uncertainty of the determined efficiency based on a 10-revolution averaged measurement.

## 6.2 Operational curve test

In the second test, the rotational speed followed a 27-step profile up to the rated speed, while the torque was kept at the nominal level of the TTS, namely 5 MN·m. Figure 8 shows the actual test progress. Since only a limited number of revolutions were available on each step, the efficiency shown in Figure 9 was calculated with the average of just one revolution, denoted as $\eta_1$. For each step, calculation was done based on the data from six revolutions, so six efficiency points, each representing the average of a single revolution, are shown. Within each step, the deviation of the six $\eta_1$ points is clearly shown. The standard deviation $\sigma_{\eta_1}$ for each test step can be calculated using the corresponding six points.

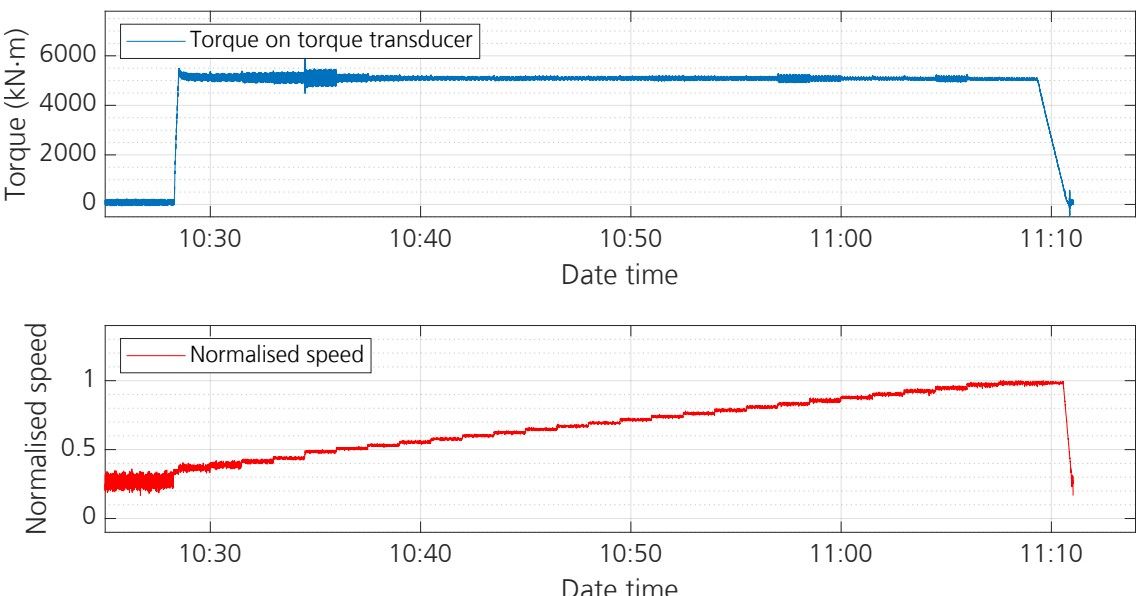

**Figure 8.** Actual test progress of the operational curve test

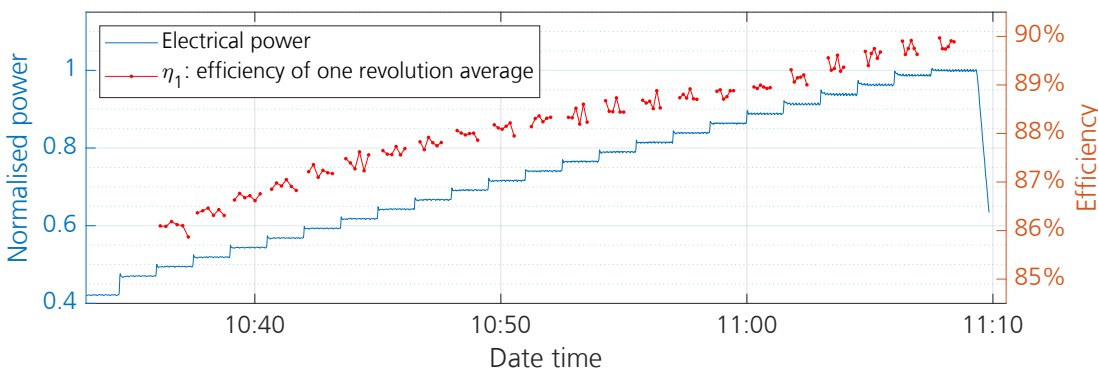

**Figure 9.** Efficiency of one revolution average for the operational test

To obtain the efficiency of each test step, the measurements of all six revolutions were used to determine the six-revolution averaged efficiency $\eta_6$. As pointed out by Song et al. (2023), the measurements of at least six full revolutions should be averaged to achieve a good level of accuracy. The results of $\sigma_{\eta_6}$ for some of the test steps are listed in Table 5. Since there were not enough revolutions to determine the standard deviation $\sigma_{\eta_6}$, the value of $\sigma_{\eta_1}$ is used instead for the calculation of the uncertainty. It is worth pointing out here that the standard deviation of a single revolution's average efficiency is adopted directly instead of in a form similar to Equation 10. This is because six points represent a very limited basis to obtain a reliable

calculation of $\sigma_{\eta_1}$. Using $\sigma_{\eta_1}$ directly as $\sigma_{\eta_6}$ serves to yield conservative results in the uncertainty analysis. The overall uncertainty of the determined efficiency is also given in Table 5. Since the torque remains at 5 MN·m throughout all the test

steps, the uncertainty due to the measurement chains is identical to the value in Table 4 for all the steps: $u_{\eta,\text{meas}} = 0.35\%$. This is therefore not listed again in Table 5. The results show that $u_{\eta,\text{meas}}$ plays a dominant role in the uncertainty of the efficiency.

**Table 5.** Determined efficiency and its uncertainty of some of the test steps

| Step | 7 | 11 | 15 | 19 | 23 | 27 |
|---|---|---|---|---|---|---|
| $n$ (normalised) | 0.54 | 0.64 | 0.73 | 0.82 | 0.92 | 1.00 |
| $\eta_6$ | 86.38% | 87.43% | 88.12% | 88.65% | 89.15% | 89.84% |
| $u_{\text{ins}} = \sigma_{\eta_1}$ | 0.06% | 0.16% | 0.09% | 0.13% | 0.11% | 0.09% |
| $U_{\eta_6}(k=2)$ | 0.72% | 0.76% | 0.72% | 0.74% | 0.74% | 0.72% |

## 7 Discussion

The efficiency determination for both of the tests discussed above achieved an accuracy of approximately 0.7% uncertainty, thereby breaking the 1% mark. The largest uncertainty contribution still comes from the torque measurement, despite the use of the best possible torque transducer and calibration machine. To further reduce the uncertainty, the transducer needs to be calibrated to a higher level of torque. PTB is commissioning a new torque calibration machine with a capacity of 5 MN·m, and this could help achieve better uncertainty.

The second largest contribution comes from the voltage measurement. Although in this case it stems from safety regulation requirements and could in particular circumstances be solved by a dedicated reconfiguration of the test bench, it still shows the importance of planning effort and investment in electrical power measurement. In practice, it should not be taken for granted that electrical power can be measured automatically with sufficient accuracy. Because testing time on a nacelle test bench is a limited resource (drivetrain efficiency would very likely be tested together with many other test items), it is not always possible to reconfigure the test layout just for one test. Therefore, it is important to plan the test in advance and take all relevant factors into consideration in order to achieve the best possible electrical measurement accuracy.

Rotational speed and electrical current measurements achieved very high levels of accuracy. For these two cases, suitable sensors with careful calibration were instrumented at the right positions on the drivetrain and integrated into well calibrated measurement chains. All these factors combined produce satisfying results.

Owing to a number of discussed reasons, ripples on the measurement and deviations in the determined efficiency are inevitable. To achieve stable efficiency under certain conditions, it is recommended to average at least six full revolutions of measurement. Based on the results of this study, the uncertainty caused by the deviation in efficiency will only represent a minor contributor to the overall MU of efficiency if this recommendation is followed.

One limitation of the test layout presented in this paper is that the non-torque loads, such as bending moments and shear forces, could not be applied to the DUT because the 5 MN·m TTS from PTB is not designed to withstand high levels of non-torque loads. To overcome this limitation, a series of calibration profiles was carried out during the test campaign so that the DyNaLab transducer developed in-house at Fraunhofer IWES could be calibrated. This transducer was placed directly in front

of the reference transducer from PTB. The torque calibration of this transducer has been reported by Zhang et al. (2023). This transducer is designed to withstand and measure loads in all six degrees of freedom and will be used for torque measurement in future test campaigns.

## 8 Conclusions

This paper reported an approach to determine the drivetrain efficiency of a modern multi-megawatt wind turbine drivetrain. Addressing the challenge of measurement accuracy, state-of-the-art sensors, measurement systems, and calibration facilities were employed within the framework of the WindEFCY project. The results show that an overall MU of 0.7% is achievable for an efficiency determination with torque measurement up to 5 MN·m. As expected, torque measurement contributed the largest share to the overall MU. Surprisingly, the electrical power measurement also played a significant role in the MU. This highlights the fact that although electrical measurements are generally considered to be much more accurate, equal care must be exercised when measuring both electrical and mechanical power. Speed measurement using an inclinometer, however, yielded very good results including a very small MU of 0.01%. To achieve a stable efficiency result, the measurement of at least six full revolutions was averaged, resulting in nearly negligible contributions to the overall MU.

. HZ concepted the paper, carried out the data analysis and wrote the major part of the paper. PW and ZS were responsible for torque and speed measurement, as well as the corresponding text in this paper. CM and AD were responsible for electrical power measurement, as well as the corresponding text in this paper. AD also carried out the comprehensive calibration of the HILO voltage dividers. MH and KE coordinated the project and test execution, BT took part in the sensor instrumentation.

. No competing interests exist.

. The project 19ENG08 – WindEFCY has received funding from the EMPIR programme co-financed by the Participating States from the European Union's Horizon 2020 research and innovation programme. The input of all the project partners is gratefully acknowledged.

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
