# Peer review of "State-of-the-art efficiency determination of a wind turbine drivetrain on a nacelle test bench"

_Wind Energy Science, 2024_

## Author Response (AR1)

**Comments of Reviewer 1**

**Comment:**
The angular speeds applied during the tests are not mentioned in the text, but are identified as normalized. It would be good to have the nominal speeds (in rpm or/and rad/s) so that the reader can have an idea of the dynamics that are being addressed here.

**Author's response**
Since the rated speed of the wind turbine is strongly depended on the rotor diameter, it is decided after discussion with the turbine manufacturer to normalize the rotational speed in the paper.

**Author's changes**
The background why the speed is normalised is explained in page 12 line 263
* * *
**Comment:**
In the abstract: Why focus on torque in the abstract, since other quantities are involved in the research and also generated important results?

**Author's response**
Instead focusing on the torque accuracy, the accuracy of both mechanical and electrical power is named as challenges for the efficiency determination

**Author's changes**
Page 1 line 4
* * *
**Comment:**
Improve the introduction to the WindEFCY project, since it is a kind of main reference to the article.

**Author's response**
The introduction of the WindEFCY project is given in the chapter "Background"

**Author's changes**
Page 3 line 3
* * *
**Comment:**
In Section 4.1, the TTS is described as "manufactured by HBM and acquired by PTB". Harmonize the information that is contradictory to Section 1, where it is mentioned as "developed by PTB".

**Author's response**
The text was corrected accordingly. The 5 MN m torque transducer was designed and manufactured by HBK (HBM at that time) for static torque measurement. PTB developed a transportable data acquisition system including a stand-alone data transmission for the application of the torque transducer under rotation. By calibrating and analysing the torque transducer, it was established as a torque transfer standard. Moreover, within the WindEFCY project, the transducer was equipped with an inclinometer to measure rotational speed. By calibrating the inclinometer and combining torque and rotational

speed measurement in one device, a new transfer standard for mechanical power measurement was formed.

**Author's changes**
Page 2 line 46, it is change to "owned" by PTB
* * *
**Comment:**
- In Section 4.1, text between lines 79 and 98: This whole procedure is very important to define the existing torque traceability. The suggestion is to open the description with more details, referring to previous studies that allowed this one to consider the extrapolation method as a consolidated approach.
- The expression of the uncertainty values should be more detailed with regard to the coverage factor, for example. Elements of DIN 51309, considered for the uncertainty assessment, should be better introduced, such as the classification parameters and the calibration case.

**Author's response**
The extrapolation of the torque calibration as well as the uncertainty determination of the TTS is comprehensively demonstrated in section 4.1 now.

**Author's changes**
Page 6 and 7
* * *
**Comment:**
- Lines 124 and 125: I think that the concept to be achieved here is that of a Traceable Mechanical Power measurement, since there is torque and speed measured with known MUs.

Suggestion to calculate the combined uncertainties to find the "Pmech MU", as is done for the "Pelec MU"

**Author's response**
Rotational speed measurement is replaced with mechanical power measurement

**Author's changes**
Page 9 line 200
* * *
**Comment:**
- It is worth having a brief description of the acquisition system configuration. This is fundamental to understand the dynamic behaviors versus the static calibration chain. In addition, some minimal information about the calibrations should be harmonized (e.g.: Line 112 contains a comment about the digital filter used for angle calibration, but for torque this parameter is not mentioned)

**Author's response**
It is a very good point to note the difference between static calibration and the dynamic testing. The electrical power is measured and calculated by the power analyser, which takes into account of the dynamic behaviours of the high-frequency sampled voltage and current signals. More description of the electrical measurement is given in chapter 5. For the measurement of mechanical power, since the system has a low rotational speed and stable torque level, it's not a highly dynamic system. The sampling rate of the torque and

speed measurement signals are considered sufficient to capture the dynamics in mechanical power.

**Author's changes**
Page 9 line 203 - 206
* * *
**Comment:**
- Lines 190 and 191: To enrich the article, it is important to look for sources of uncertainty within the 10-revolution package drive. In addition, the amplitude of the variation is important due to mechanical stresses and metrological parameters. One suggestion is to deepen the relationships between the evaluated hypothesis and the metrological parameters (e.g.: under a torque overshoot, can the transducer show any hysteresis? with speed variations, is there any considerable acceleration for the shaft?)

**Author's response**
Thank you very much for the input of new ideas for future work. It would be very interesting to check the effect of following events in during the test: 1) overshoots in the torque and 2) Overshoots in the speed. During the test described in this paper, the speed and torque were held stable, which corresponds to the aim to determine the efficiency at a stable state. But in the field, the turbines are constantly subjected to variable wind conditions. For further investigations it is very interesting to know what dynamic behaviours in speed and power influences the overall efficiency of the drivetrain.

**Author's changes**
None
* * *
**Comment:**
- In the Section Discussion, the text within Lines 258 to 264 is not clear. It is as if there were two PTB torque transducers. Is this correct?

**Author's response**
It was a mistake in the text. The transducer to be calibrated is made in-house at Fraunhofer IWES, not PTB. This has been corrected.

**Author's changes**
Corrected in page 17 line 350
* * *
**Comment:**
- Open some acronym explanation as the first time they are used (e.g.: PTB, WindEFCY, MU)

**Author's response**
Addressed in the new version

**Author's changes**
Page 2 line 47
Page 3 line 3 line 66
* * *
**Comment:**
- Correct the description of PTB to National Metrology Institute, not Metrological...

**Author's response**
Addressed in the new version

**Author's changes**
Page 2 line 47
* * *
**Comment:**
- Revise Figure 1 to include identification of all relevant components, e.g., LAU and DUT are not identified

**Author's response**
Addressed in the new version

**Author's changes**
Page 4 Figure 1
* * *
**Comment:**
- Equation 1, there is no introduction to the variable Pmech in the text

**Author's response**
Addressed in the new version

**Author's changes**
Page 4 line 99
* * *
**Comment:**
- In Figure 2, identify and detail the components of the left and right image.

**Author's response**
Figure improved

**Author's changes**
Page 5 Figure 2, more description in the caption is added
* * *
**Comment:**
- Do not break lines to separate the units of their values (e.g.: in Section 4.1 "1.1 MN m")

**Author's response**
Addressed in the new version

**Author's changes**
Corrected in LaTeX code so that no break line happens in the middle of the units

**Comment:**
- In Table 2, last column, the parameter is not "Pelec", but "Pelec MU"

**Author's response**
The caption of the table is extended. The content of the whole table is about the measurement uncertainty of different variables; therefore, this is directly stated in the caption.

**Author's changes**
Page 11 Table 2
* * *
**Comment:**
- Revise and improve the detail of the captions for figures and tables

**Author's response**
This is carried out for multiple figures and tables

**Author's changes**
Captions of multiple tables and figures are changes with more detailed information
* * *
**Comment:**
- Figure 6, there is no unit for Power

**Author's response**
Due to the same reason of the normalised rotational speed, the power also needs to be normalised.

**Author's changes**
No direct change made addressing the Figure6, but it is explained direct earlier in the text why the speed is normalised

**Comments of Reviewer 2**

**Comment:**
The title "state of art efficiency determination" is misleading. It is difficult to tell whether the presented efficiency measurement method is state of art, without having a thorough literature review.

**Author's response**
This contribution describes results of the WindEFCY project, officially titled "Traceable mechanical and electrical power measurement for efficiency determination of wind turbines", that was a collaborative research initiative across disciplines under the European Metrology Programme for Innovation and Research (EMPIR). Aim of the project was the development and validation of standardised test methods for the efficiency determination of wind turbines and their components on test benches in a reliable, reproducible, and comparable way for quality assurance. In chapter 2 we are now giving more information about this background.

**Author's changes**
More information given in chapter 2 (page3)
* * *
**Comment:**
Literature review: efficiency measurement is commonly done on a laboratory for a nacelle component, such as gearbox or generator. What are the various resources of efficiency loss have not discussed? Without knowing this, little knowledge can be gained to improve efficiency in design

**Author's response**
The focus of our work was on reproducible measurements for subsequent use to quantify measures to minimise losses. We have now included more explanation on calibration and traceability and why this is so relevant for future work to improve efficiency.

**Author's changes**
Page 2 line 28 to line 49
* * *
**Comment:**
Results: a limited set of data has been presented for the nacelle efficiency. As wind turbine is operation at a wide range of wind conditions. More results are desired to have an overview of nacelle efficiency range during operations

**Author's response**
In the WindEFCY project, we used the world's largest torque transducer (at the time) with a range of up to 5 MNm. Unfortunately, this limited our testing capabilities as we were not able to analyse a wide range of wind conditions. However, as our focus was on reproducible measurements, this was an acceptable limitation for our work.

**Author's changes**
Unfortunately, the scope of the results in this paper could not be more extended

**Comment:**
Overall writing can be improved as well.

**Author's response**
This has been improved in the new version.

**Author's changes**
Various positions. Please refer to the change-tracking document